# QUDEVAL: The Evaluation of Questions Under Discussion Discourse Parsing

**Yating Wu**[1]   **Ritika Mangla**[2]   **Greg Durrett**[2]   **Junyi Jessy Li**[3]
[1]Electrical and Computer Engineering, [2]Computer Science, [3]Linguistics
The University of Texas at Austin
{yating.wu, ritikamangla, gdurrett, jessy}@utexas.edu

## Abstract

Questions Under Discussion (QUD) is a versatile linguistic framework in which discourse progresses as continuously asking questions and answering them. Automatic parsing of a discourse to produce a QUD structure thus entails a complex question generation task: given a document and an answer sentence, generate a question that satisfies linguistic constraints of QUD and can be grounded in an anchor sentence in prior context. These questions are known to be curiosity-driven and open-ended. This work introduces the first framework for the automatic evaluation of QUD parsing, instantiating the theoretical constraints of QUD in a concrete protocol. We present QUDEVAL, a dataset of fine-grained evaluation of 2,190 QUD questions generated from both fine-tuned systems and LLMs. Using QUDEVAL, we show that satisfying all constraints of QUD is still challenging for modern LLMs, and that existing evaluation metrics poorly approximate parser quality. Encouragingly, human-authored QUDs are scored highly by our human evaluators, suggesting that there is headroom for further progress on language modeling to improve both QUD parsing and QUD evaluation.

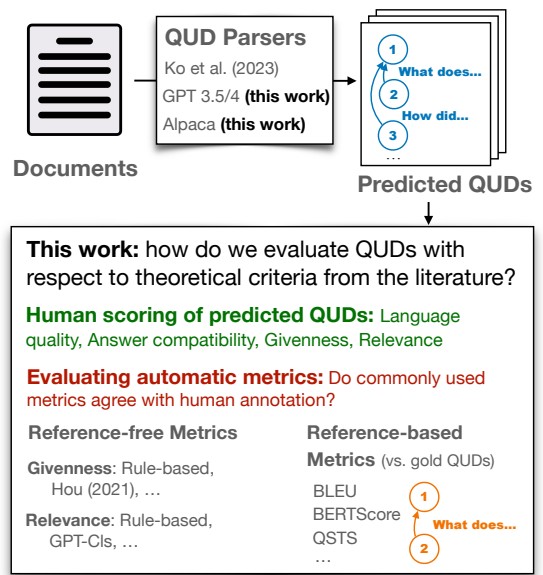

Figure 1: An overview of this work. Given QUD parser outputs, we (a) design a protocol for evaluation; (b) collect human judgments using this protocol; (c) evaluate automatic evaluation metrics. This work additionally contributes new LLM-based QUD parsers.

## 1   Introduction

Text comprehension at the discourse level entails understanding higher level structures between sentences and paragraphs, beyond the meaning of individual words. The linguistic framework of Questions Under Discussion (QUD) (Van Kuppevelt, 1995; Roberts, 2012; Benz and Jasinskaja, 2017) views the progression of discourse as continuously posing (implicit) questions (i.e., QUDs) and answering them; thus each sentence in a monologue is an answer, or part of an answer, to a QUD. In Figure 2, the third sentence answers the implicit QUD, "*What does Glenn say about the situation?*", elicited from sentence 2. The advent of large language models (LLMs) makes viewing discourse in a question generation and answering fashion increasingly

tractable for settings that require higher-level processes, e.g., planning in text generation (Narayan et al., 2023), contextualization in question answering (Newman et al., 2023), and elaborative simplification (Wu et al., 2023). However, rigorous evaluation of the question generation aspect of QUD frameworks remains an open challenge.

QUD parsing is relatively new to the NLP community, with most of the prior work in discourse parsing focusing on structures like Rhetorical Structure Theory (Mann and Thompson, 1988), Segmented Discourse Representation Theory (Asher et al., 2003), and the Penn Discourse Treebank (Prasad et al., 2008), all of which use a fixed, hierarchical discourse relation taxonomy and are relatively straightforward to evaluate using accuracies or F measures. QUD parsing (illustrated in Figure 2), however, entails a question *generation* task that is contextually grounded (Ko et al., 2023).

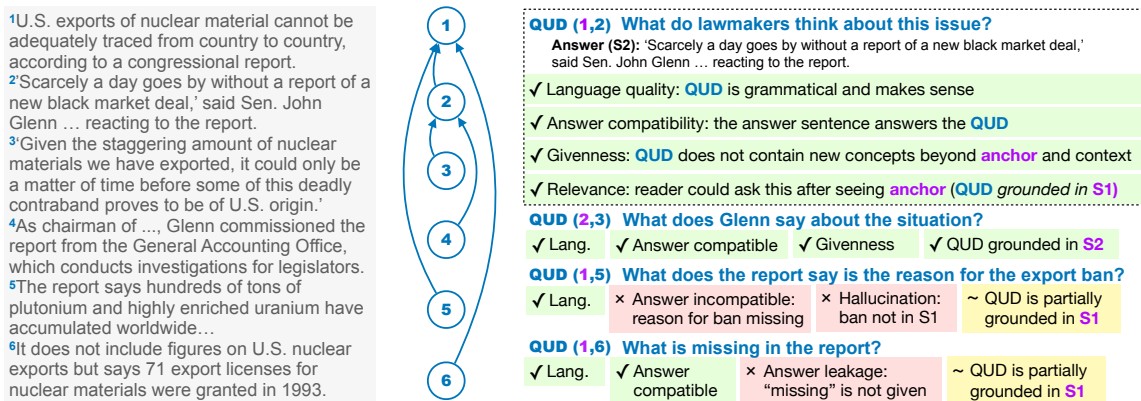

Figure 2: A QUD dependency parse and our evaluation labels; showing 4 of the 5 QUDs due to space constraints. Given the article snippet on the left, a QUD dependency parser generates the tree with edges as QUDs; QUD(2, 3) means that sentence 3 answers the corresponding question anchored in sentence 2. Besides generating the question, the position of the anchor sentence (head node) is also predicted. QUDEVAL includes 4 criteria for each edge (right) evaluating both the generated QUD and the predicted anchor, discussed in Section 3.

This structure necessitates two key principles for evaluation. First, the questions are raised after a certain amount of common ground is established, so parsing QUD involves identifying where in the context a question *can* be triggered (also called the *anchor sentence*). These anchors are produced by parsers and form part of the structure to evaluate. Second, for a question to become a QUD, it needs to satisfy several linguistic constraints (Riester et al., 2018) and be plausible *without* knowing a priori how the discourse will progress.

The above characteristics make QUD parsing much trickier to evaluate than other discourse parses; thus, prior work in QUD dependency parsing (Ko et al., 2023) performed only manual evaluation. Automatic evaluation for generation tasks is known to be challenging (Celikyilmaz et al., 2020) and standard evaluation metrics for question generation (Amidei et al., 2018; Nema and Khapra, 2018; Gollapalli and Ng, 2022) tend to be superficial linguistically. They do not evaluate the aforementioned desiderata of QUD.

Outlined in Figure 1, this work provides foundational building blocks for the automatic evaluation of QUD dependency parsers. Building on prior theoretical work on QUD and its annotation (Riester et al., 2018; De Kuthy et al., 2018), as well as error analysis on a large number of machine outputs from several different models, we design four key criteria to assess QUD question generation quality, covering language quality, answer compatibility, question givenness, and anchor relevance.

With these criteria, we present QUDEVAL, 2,190 generated question-anchor pairs annotated by trained linguists across 51 news articles. These annotations evaluate three QUD dependency parsing models: Ko et al. (2023)'s pipeline system, as well as our newly designed prompts for QUD parsing with LLMs using OpenAI ChatGPT (GPT 3.5 Turbo), GPT-4, and Stanford Alpaca (Taori et al., 2023). Human annotations using the QUDEVAL framework reveal that modern LLMs, while able to generate fluent questions, often fail at satisfying these constraints; the errors vary across systems, rendering such fine-grained evaluation necessary.

QUDEVAL also provides a valuable testbed for the assessment of automatic evaluation metrics for QUD parsers. We engage with rule-based and LLM-based baselines, prior work relevant for specific criteria, and reference-based metrics standard in question generation evaluation. Results show that while these metrics align with QUDEVAL's annotations to some extent, they give poor estimates of QUD generation quality (except for anchor groundedness) in comparison to an estimated human upper bound. This points to both challenges and substantial headroom for the development of automatic metrics.

To sum, QUDEVAL points to clear directions for future progress: (1) human evaluation reveals consistently higher quality for crowdsourced QUDs compared to generated ones, suggesting opportunities for better QUD parsers; (2) the development of linguistically-informed automatic metrics targeting QUD parsing quality is necessary, for which QUDEVAL can serve as a benchmark.

QUDEVAL is available at https://github.com/lingchensanwen/QUDeval.

## 2 Background and Related Work

**QUD Annotation and Parsing** Despite its rich linguistic history (see Benz and Jasinskaja (2017) for an overview), QUD's presence in NLP is in its infancy as annotated datasets have only recently started emerging. These datasets follow different QUD paradigms, reflecting its diverse interpretation: Westera et al. (2020) and Ko et al. (2020) presented QUDs in an expectation-driven (Kehler and Rohde, 2017) manner, where questions are elicited *while* reading (i.e., without seeing the rest of the article). Unanswerable questions are thus a natural artifact of this process. De Kuthy et al. (2018) annotated QUD trees with full, hierarchical questions (i.e., the answer to a parent question entails the answer to a child question (Roberts, 2012)); however, due to the challenging annotation process, the dataset only contains two sections of an interview transcript, too small to train automatic parsers.

The parsing framework that this paper engages with is from Ko et al. (2023), trained with the DCQA dataset from Ko et al. (2022). Ko et al. (2023) views the QUD structure as a *dependency tree*, where each sentence in a document is connected to an anchor sentence in its prior context via a QUD (formalized in Section 3). This is a lightweight approach to QUD as the questions themselves are not guaranteed to be hierarchical; however this simplification allows large-scale, crowdsourced data collection (Ko et al., 2022). To date, this remains the only QUD parser available.

**Evaluation of Question Generation** Prior work has tackled contextual question generation, including conversational (Nakanishi et al., 2019; Gu et al., 2021; Do et al., 2022; Kim et al., 2022), open-ended (Ko et al., 2020; Cao and Wang, 2021; Chakrabarty et al., 2022), and nested questions (Krishna and Iyyer, 2019). While some of these works perform human evaluation, they typically cover dimensions such as fluency, relevance/plausibility, scope, and question-answer consistency. In clarification question generation (Rao and Daumé III, 2019), dimensions such as usefulness and informativeness are also considered. However, the above criteria are insufficient for a linguistically meaningful evaluation of QUD generation. Ko et al. (2023) used a human evaluation taxonomy for their QUD parser; this work reflects an extended version of that earlier taxonomy, better aligned with theoretical principles.

Similar to automatic evaluation of Natural Language Generation (NLG) tasks in general (Celikyilmaz et al., 2020), automatic evaluation of question generation is known to be challenging and inconsistent (Amidei et al., 2018). We evaluate commonly-used metrics, as well as a recent metric from Gollapalli and Ng (2022), using our linguistically-grounded frameworks.

## 3 QUD Evaluation Framework

QUDs are subjective: different readers often come up with distinct questions even with the same answer sentence (Ko et al., 2022). Thus our goal is to target common principles that QUDs need to satisfy. First, we instantiate theoretical constraints of QUD (Riester et al., 2018; De Kuthy et al., 2018) in a concrete protocol. Second, we consider common errors in text generation systems, including question generation systems, and ensure they are captured in our taxonomies.

**Setup and Definitions** Each evaluation instance corresponds to an individual edge in a dependency tree where a QUD connects two sentences: the head of an edge is an **anchor sentence** $S_k$ (the $k$th sentence in a document) where the question is elicited, and the child node is an **answer sentence** $S_a$ that answers the question (Figure 2). Thus we denote an edge as $(Q, S_k, S_a)$ where $Q$ is the question string. QUD dependency parsing entails considering each sentence in a document as $S_a$ and, for that answer sentence, generating $\hat{Q}$ and predicting $\hat{k}$, the anchor index.

We present four criteria that assess both tasks separately and jointly, with the full annotator-facing instructions listed in Appendix Figure 4. Note that it is both theoretically and operationally possible that more than one sentence satisfies the criteria of being an anchor. In practice, we see that this depends on varying levels of specificity of the question and the document context. Because of this, our evaluation framework independently evaluates each QUD and its predicted anchor.

**Language Quality (Lang.)** This criterion is designed to filter out obviously bad questions. These include: badly formed questions that are not accommodatable, questions that are irrelevant to the article, and questions with content that obviously contradicts the article. Note that we direct annotators to *skip* the rest of the evaluation if $\hat{Q}$ fails to satisfy this criterion. Examples of such questions

are shown in Appendix Table 17.[1]

**Answer Compatibility (Comp.)** This criterion states that $S_a$ should answer $\hat{Q}$. This is one of the key criteria used both in Riester et al. (2018)'s guidelines (called "congruence"), as well as in human evaluations for QG systems, e.g., called "answer validity" in Krishna and Iyyer (2019) and "consistency" in Gu et al. (2021). We additionally consider the important role QUD plays in information structure in pragmatics (Buring, 2008; Beaver and Clark, 2009), i.e., QUDs are used to tease apart the main part (called focus or at-issue content) from background information in a sentence (Riester et al., 2018). Thus it is the *focus* of $S_a$ that should answer $Q$ (Ko et al., 2022). We introduce a graded notion of Answer Compatibility:

(1) *Direct and explicit*: full compatibility described above (examples in Figure 2).

(2) *Unfocused*: $S_a$ contains the answer, but the answer is not its focus. An example is shown in Appendix Table 18.

(3) *Not answered*: $S_a$ does not answer $\hat{Q}$. In Figure 2, sentence 5 should be the answer for the generated QUD (1,5); however, sentence 5 does not state the reason for the export ban. Another example is shown in Figure 3.

**Givenness (Givn.)** Since QUDs should be reader questions invoked at $S_{\hat{k}}$ for upcoming, unseen discourse given the common ground already established, $\hat{Q}$ should only contain concepts that are *accessible* by the reader from prior context; this is called "Q-Givenness" in Riester et al. (2018). While they loosely define givenness as "given (or, at least, highly salient) material", we concretely specify this notion with information status (Markert et al., 2012): $\hat{Q}$ should only contain concepts (entities, events, or states) that were mentioned in the question context $C_{\hat{Q}} = S_1, ..., S_{\hat{k}}$ (discourse-old), or concepts that have not been directly mentioned but are generally known or inferrable from mentioned ones (mediated).

Based on observations of machine errors from Ko et al. (2023), generated questions can often contain concepts that are from $S_a$ itself, which is a particular error that prevents a QG model being used more widely in conversations (Nakanishi et al., 2019). We call this answer leakage. This criterion is divided into the following categories:

(1) *No new concepts*: all concepts in $\hat{Q}$ are discourse-old or mediated (examples in Figure 2).

(2) *Answer leakage*: $\hat{Q}$ contains new concepts not in question context $C_{\hat{Q}} = S_1, ..., S_{\hat{k}}$ but in $S_a$. In Figure 2, QUD (1,6) states "*What is missing in the report?*"; yet the notion of "missing" is absent in $S_{\hat{k}}$ (S1) and only introduced in $S_a$ (S6), i.e., based on the common ground established in $S_{\hat{k}}$, it is quite a stretch for a reader to ask this question. Figure 3 shows another example.

(3) *Hallucination*: $\hat{Q}$ contains new concepts that are not answer-leakage. An example is shown in Appendix Table 19.

Note that since $S_{\hat{k}}$'s position $\hat{k}$ is predicted, this criterion jointly evaluates $\hat{Q}$ and $\hat{k}$.

**Anchor Relevance (Relv.)** Our last criterion evaluates the anchor prediction $\hat{k}$ given $\hat{Q}$. This diverges significantly from Riester et al. (2018) since anchor prediction is not a task in human annotation. Yet, we align with their work stating that $\hat{Q}$ should be *relevant* (i.e., contains given material) to the context where it was elicited. Referencing observations from our experiments and Ko et al. (2023), we design three categories:

(1) *Fully grounded*: $\hat{k}$ satisfies the desiderata above, which typically means that content in $\hat{Q}$ follows mostlyfrom $S_{\hat{k}}$ (examples in Figure 2).

(2) *Partially grounded*: some content from $\hat{Q}$ is grounded in $S_{\hat{k}}$. For QUDs (1,5) and (1,6) in Figure 2, $S_{\hat{k}}$ (S1) only contain some of the concepts accessible from $\hat{Q}$.

(3) *Not grounded*: content from $\hat{Q}$ is largely not from $S_{\hat{k}}$. For the ChatGPT-generated question in Figure 3, the concept of "restrictions" is the main focus of the question, which is irrelevant to $S_{\hat{k}}$ (S5) that provides more information on the contents of the report.

# 4 The QUDEVAL Dataset

This section describes our linguist-annotated QUDEVAL dataset evaluating fine-tuned (Ko et al., 2023) and LLM parsers, and their results.

## 4.1 Parsers Evaluated

**Ko et al. (2023)** This parser is trained as a pipeline, with a Longformer model (Beltagy et al., 2020) for anchor prediction and GPT-2 (Radford et al., 2019) for question generation; both are fine-tuned on the DCQA dataset (Ko et al., 2022).

**LLMs** We also experiment with **ChatGPT** (GPT 3.5 Turbo), as well as Stanford **Alpaca**-7B (Taori

---

[1] For wrong predictions of $\hat{k} = a$, the annotators also skip the QUD; this happens in about 3% of the LLM parsers and does not happen with Ko et al. (2023)'s parser.

| | Lang. | | Comp. | | | Givn. | | | Relv. | | |
|---|---|---|---|---|---|---|---|---|---|---|---|
| | Yes | No | Dir-Ans. | Unfocus. | Not-Ans. | No-New. | Ans-leak. | Hallu. | Full. | Part. | No-G. |
| Ko et al. | 92.5 | 7.5 | 52.5 | 11.0 | 36.5 | **75.6** | **12.1** | 12.3 | **70.1** | **19.3** | 10.6 |
| ChatGPT | 96.1 | 3.9 | 82.4 | 8.8 | 8.8 | 64.9 | 31.2 | **3.9** | 56.7 | 31.6 | 11.6 |
| Alpaca | 93.9 | 6.1 | 43.2 | 16.9 | 39.9 | 61.2 | 30.7 | 8.1 | 46.3 | 25.5 | 28.2 |
| GPT4 | **100.0** | **0.0** | **90.6** | **3.9** | **5.5** | 61.8 | 34.3 | **3.9** | 54.1 | 35.5 | **10.4** |
| DCQA | 98.0 | 2.0 | 71.4 | 16.4 | 12.2 | 85.7 | 10.9 | 3.4 | 80.3 | 17.7 | 2.0 |

Table 1: Distribution (in %) of human-annotated labels for 2,040 system-generated QUDs and 150 crowdsourced QUDs in DCQA. For Comp., Givn. and Relv., these percentages are calculated over the total # of questions that passed the Language Quality criteria. The error distributions (other than Lang.) across models are significantly different from each other (Chi-Square, $p < 0.05$) except for ChatGPT vs. GPT-4 for Givn. and Relv.

et al., 2023), as LLM-based parsers.[2] We first prompt the model to generate $\hat{Q}$ given the article and $S_a$. Subsequently, we used $\hat{Q}$, the article, and $S_a$ to generate the $S_{\hat{k}}$. We experimented with various zero-shot and few-shot variations; our best prompt containing four in-context examples is provided in Table 8 in the Appendix. In particular, we explicitly prompted the model not to introduce phrases or ideas in the question which are newly introduced in $S_a$.

Finally, we collected QUDs with **GPT-4** after its API became available. This portion of the data (510 QUDs) was triply annotated as an addition to the dataset. However, due to the lack of API access at the time of experimentation, this portion of the data was not part of *metric* evaluation in Section 5.

## 4.2 Human Data Collection

**Data Sourcing** We run the above parsers on all news articles from the validation and test sets from Ko et al. (2023), which came from the DCQA dataset (Ko et al., 2022). For each document, we sample 10 sentences as the set of answer sentences; for each answer sentence, 4 distinct $(\hat{Q}, S_{\hat{k}})$ pairs were generated, one from each system. This amounts to 2,040 *machine-generated* QUDs in total. Additionally, we annotated 150 *crowdsourced* questions in DCQA across 15 articles both as an assessment of prior annotated QUD questions and a validation of our taxonomy. Thus the total number of distinct QUDs annotated is 2,190.

**Annotation Procedure and Agreement** Our annotation team consists of 3 students in a linguistics department; these students had extensive prior experience in data annotation before this task. The students were trained with our annotation guide-

---

[2]A temperature of 0 is used throughout the paper for LLMs. We noticed that increasing the temperature for our setting resulted in the generated questions and anchors containing concepts which are outside of the context provided.

| | Lang. | Comp. | Givn. | Relv. |
|---|---|---|---|---|
| **Krippendorff's $\alpha$** | 0.56 | 0.52 | 0.48 | 0.51 |
| **3/3 Agr %** | 0.91 | 0.58 | 0.64 | 0.57 |
| **Pairwise F1** | 0.78 | 0.61 | 0.60 | 0.60 |

Table 2: Krippendorff's $\alpha$ (nominal for *Lang.* & *Givn.*, ordinal for *Comp.* & *Relv.*); % of total agreement; pairwise F1 scores macro-averaged across labels.

lines on 20 questions (2 news articles × 2 systems) our team has iterated on. We release our **annotation interface**, presented in Appendix A.

Next, all three annotators annotated 200 distinct QUDs (10 articles, 10 answer sentences each, and 2 QUDs for each answer sentence). This set is used to calculate inter-annotator agreement, reported in Table 2. Krippendorff's $\alpha$ (Krippendorff, 2011) values indicate moderate agreement (Artstein and Poesio, 2008) across all criteria. Table 2 also shows the % of questions where all 3 annotators agreed on a label, and an aggregated pairwise F1 score that reflects how accurately one annotator captures another's labels. The F1 can be viewed as a **human upper bound** for automatic metrics (Section 5).

The annotation team discussed a representative portion of the data on which they did not fully agree. In almost all cases, disagreements are genuine differences in interpretation of the article itself or borderline cases between labels; some of these examples are shown in Table 20 in the Appendix. In addition, there were 25 QUDs that involved at least one criterion without a majority label; these questions were adjudicated *after* agreement was calculated. The adjudicated labels are included in QUDEVAL.

Given (a) the high cognitive load of this task, (b) how well-trained the annotators are, and (c) the fact that we found no case of erroneous task interpretation for the triply annotated set above, the rest of the questions are annotated by one annotator.

| System | Duplicate | Avg. Len. |
|---|---|---|
| Ko et al. | 5 (1%) | 13.11 |
| Alpaca | 104 (20%) | 11.45 |
| ChatGPT | 19 (4%) | 16.88 |
| GPT4 | 16(3%) | 15.09 |

Table 3: Count and % of identical questions, as well as average number of tokens in generated questions, stratified by system. Each system generated 510 QUDs.

## 4.3 QUD Parser Evaluation Results

Table 1 shows the distribution of annotated labels for each system evaluated, and for the set of 150 DCQA questions. All models score greater than 92% in terms of Language Quality, a clear indication of their question generation capability in general. However, for all other criteria, there is a large percentage of errors. For Answer Compatibility, GPT-4 clearly leads the other models. Ko et al. (2023)'s system generates questions that are more grounded in the context, and predicts anchors that are the most relevant; however, their system scored the worst in hallucinations. None of the LLMs outperforms Ko et al. (2023)'s fine-tuned Longformer model for anchor prediction.

Notably, for all criteria other than Answer Compatibility, the models underperform crowdsourced questions in DCQA by a large margin. For Answer Compatibility, ChatGPT and GPT-4 scored higher in terms of direct answers, but we observe that this is directly linked to its tendency to leak answers in the first place. **These results show that QUD parsing remains a challenging task and that** QUDEVAL**'s evaluation framework is able to capture errors from distinct types of models.**

Qualitative analysis reveals that outputs from these models are of different styles (Figure 3). Ko et al. (2023)'s model tends to generate more *general* questions, likely resulting in fewer errors in Givenness and Anchor Relevance. On the other hand, GPT models generate more verbose questions that are too grounded in the answer sentence itself, resulting in profuse answer leakage. Table 3 shows the average lengths of questions for each system. Alpaca makes a distinct set of errors: it generates identical questions across different anchor and answer sentences up to 20% of the time, much more frequently than other models (Table 3).

## 5 Automatic Evaluation Metrics

QUDEVAL can serve as the first benchmark for automatic evaluation metrics for QUD dependency

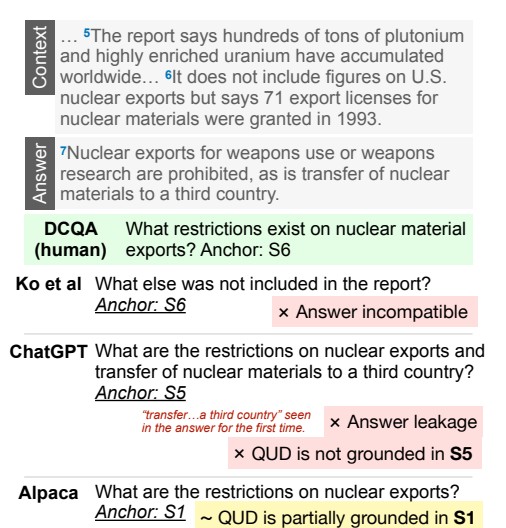

Figure 3: Model-generated questions for answer sentence 7 in the same article as Figure 2.

parsers. As a first step, we evaluate a set of baseline reference-free and reference-based metrics.[3] Note that we do *not* evaluate the Language Quality criterion, since all modern LLMs are capable of generating fluent questions (Table 1).

### 5.1 Reference-Free Metrics

Baselines tested here include rule-based metrics, prior work on information status (Hou, 2021), and classification/scoring with GPT models. The latter has shown promise for evaluation in machine translation (Kocmi and Federmann, 2023) and summarization (Luo et al., 2023; Wang et al., 2023). We held out the two articles used for annotator training as the validation set (60 questions).

**Metric Preliminaries** All metrics are classifiers taking the form $f\colon (S_a, S_{\hat{k}}, \hat{Q}, C_{\hat{Q}}) \rightarrow \mathcal{Y}$, mapping from a QUD (answer sentence, anchor sentence, predicted question, and context) to one of the labels (in the set $\mathcal{Y}$) for one of our criteria. Note that not all criteria necessarily use every piece of information; anchor relevance does not use anything about the answer sentence, for example.

We define two types of GPT4-based metrics throughout this section. First, **GPT-Cls** uses ChatGPT as either a zero-shot (**-zs**) or few-shot (**-fs**) classifier with an appropriate prompt. This directly maps into the label set $\mathcal{Y}$. Second, **GPT-Scr** follows Kocmi and Federmann (2023) and Wang et al. (2023) to use GPT-4 to assign a real-valued score between 1 and 100 for the quantity of interest. We

---

[3]As stated in Section 4.1, human evaluation results of GPT-4 generated QUDs were not included to test automatic metrics due to the lack of API access at the time of experimentation.

| Comp. | Direct Ans. | Unfocused | Not Ans. | Macro F1 |
|---|---|---|---|---|
| GPT-Scr | 0.70 | 0.05 | 0.36 | 0.37 |
| GPT-Ans | **0.84** | | **0.37** | **0.60** |
| Random | 0.62 | 0.11 | 0.31 | 0.35 |

| Givn. | No new cncpt. | Ans. leak. | Hallu-cinat. | Macro F1 |
|---|---|---|---|---|
| Rule-based | 0.52 | **0.40** | 0.19 | **0.37** |
| Hou (2021) | **0.77** | 0.09 | 0.10 | 0.32 |
| GPT-Cls-zs | 0.75 | 0.02 | **0.22** | 0.33 |
| GPT-Cls-fs | 0.61 | 0.02 | 0.19 | 0.27 |
| Random | 0.66 | 0.20 | 0.08 | 0.32 |

| Relv. | Fully grnd. | Some grnd. | Not grnd. | Macro F1 |
|---|---|---|---|---|
| Rule-based | 0.31 | 0.35 | 0.37 | 0.34 |
| GPT-Cls-zs | 0.65 | 0.34 | 0.49 | 0.49 |
| GPT-Cls-fs | 0.60 | 0.26 | 0.46 | 0.44 |
| GPT-Scr | **0.73** | **0.41** | **0.57** | **0.57** |
| BLEU1-sim | 0.59 | 0.26 | 0.22 | 0.35 |
| Random | 0.58 | 0.28 | 0.15 | 0.34 |

Table 4: Reference-free metric assessment results (in F1). For Answer Compatibility, *GPT-Ans* does not differentiate between direct answer and unfocused answer.

| Comp. | Direct Ans. | Unfocused | Not Ans. | Macro F1 |
|---|---|---|---|---|
| BLEU-1 | 0.51 | **0.14** | 0.36 | 0.32 |
| BERTScore | 0.51 | **0.14** | **0.43** | **0.36** |
| METEOR | 0.63 | 0.07 | 0.36 | **0.36** |
| ROUGE | 0.61 | 0.03 | 0.37 | 0.34 |
| QSTS | 0.52 | 0.12 | 0.38 | 0.35 |
| GPT-Scr | **0.64** | 0.08 | 0.31 | 0.35 |

| Givn. | No new cncpt. | Ans. leak. | Hallu-cinat. | Macro F1 |
|---|---|---|---|---|
| BLEU-1 | 0.61 | 0.21 | 0.12 | 0.32 |
| BERTScore | 0.4 | 0.29 | **0.13** | 0.28 |
| METEOR | 0.45 | 0.33 | 0.08 | 0.29 |
| ROUGE | **0.73** | 0.13 | 0.10 | 0.32 |
| QSTS | 0.62 | 0.30 | 0.07 | 0.33 |
| GPT-Scr | 0.65 | **0.35** | 0.1 | **0.37** |

| Relv. | Fully grnd. | Some grnd. | Not grnd. | Macro F1 |
|---|---|---|---|---|
| BLEU-1 | 0.55 | 0.19 | 0.21 | 0.32 |
| BERTScore | 0.36 | **0.27** | 0.20 | 0.28 |
| METEOR | 0.42 | **0.27** | **0.24** | 0.32 |
| ROUGE | **0.63** | 0.19 | 0.18 | 0.34 |
| QSTS | 0.57 | 0.21 | 0.22 | 0.34 |
| GPT-Scr | **0.63** | 0.26 | 0.22 | **0.37** |

Table 5: Reference-based metric assessment results.

then use a **mapping function** to convert values in this interval to labels in $\mathcal{Y}$, tuned for macro-F1 on the validation set. Such a mapping from this interval is possible because many of our error categories have an ordinal semantics associated with them.

**Answer Compatibility** (1) *GPT-Scr* assesses how well $S_a$ answers $\hat{Q}$. Mapping function and prompt detailed in Appendix B. (2) *GPT-Ans*: Wadhwa et al. (2023) showed the efficacy of GPT-4 on a QA setting similar to QUD. We prompt GPT-4 using their prompt to generate an answer, given the article, $\hat{Q}$, $S_{\hat{k}}$. We then prompt it again to find the closest sentence in the article to the generated answer (prompt in Appendix Table 10).

**Givenness** (1) *Rule-based*: We scan lemmatized content words in $\hat{Q}$ for new words not present in $C_{\hat{Q}}$; if they appear only in $S_a$, then we label $Q$ as 'answer leakage', otherwise as 'hallucination'. (2) *Hou (2021)*: we run the state-of-the-art information status classification model from Hou (2021), using the top-level labels (new, old, mediated). Since both new and mediated concepts are allowed, we merge these two classes. We use a similar rule as (1) to differentiate between answer leakage and hallucination, detailed in Appendix B. (3) *GPT-Cls*: the prompts can be found in Appendix Tables 12 (zero-shot) and 13 (few-shot).

**Anchor Relevance** (1) *Rule-based*: This method checks if the focus (approximated by maximum NP) of $\hat{Q}$ overlaps with the predicted anchor $S_{\hat{k}}$.

We check for content word overlap with the $S_{\hat{k}}$. $\hat{Q}$ is labeled as 'fully grounded' if all words in its max NP are present in $S_{\hat{k}}$, 'not grounded' if none are present, and 'partially grounded' otherwise. (2) *GPT-Cls*: the prompts are in Appendix Tables 14 (zero-shot) and 15 (few-shot). (3) *GPT-Scr*: mapping function and prompt are in Appendix B. (4) *BLEU1-sim* scores an answer by computing BLEU-1 (Papineni et al., 2002) between $\hat{Q}$ and $S_{\hat{k}}$; mapping function detailed in Appendix B.

## 5.2 Reference-Based Metrics

We further evaluate reference-based metrics standard in question generation that capture the similarity between $\hat{Q}$ and $Q$: (1) **BLEU-1** (Papineni et al., 2002);[4] (2) **BERTScore** (rescaled) (Zhang et al., 2020); (3) **METEOR** (Lavie and Agarwal, 2007); (4) **ROUGE**-F1 (Lin, 2004); (5) **QSTS** (Gollapalli and Ng, 2022), a recent reference-based metric for question generation evaluation that explicitly represents the question class and named entities in a given question pair, and combines them with dependency tree information and word embeddings to measure the semantic similarity between two questions;[5] (6) **GPT-Scr** (prompt in Appendix Ta-

---

[4]While it is standard to report BLEU 1, 2, 3, 4, Nema and Khapra (2018) found that BLEU-1 correlates with human annotations better than other settings.

[5]The original QSTS uses the harmonic mean to combine several sub-scores, which yields a score of zero when there

|  |  | Comp. | | | Givn. | | | Relv. | | |
| --- | --- | --- | --- | --- | --- | --- | --- | --- | --- | --- |
|  |  | Dir-Ans. | Unfocus. | Not-Ans. | No-New. | Ans-leak. | Hallu. | Full. | Some. | No-G. |
| **Reference Free** | Ko et al. | 0.88 | | 0.11 | 0.82 | 0.09 | 0.09 | 0.75 | 0.33 | 0.46 |
| | ChatGPT | 0.75 | | 0.24 | 0.86 | 0.10 | 0.04 | 0.75 | 0.50 | 0.51 |
| | Alpaca | 0.96 | | 0.03 | 0.72 | 0.10 | 0.18 | 0.68 | 0.36 | 0.65 |
| | DCQA | 0.82 | | 0.17 | 0.76 | 0.15 | 0.09 | 0.71 | 0.39 | 0.14 |
| **Reference Based** | Ko et al. | 0.04 | 0.03 | 0.92 | 0.29 | 0.49 | 0.21 | 0.29 | 0.24 | 0.46 |
| | ChatGPT | 0.50 | 0.13 | 0.36 | 1.0 | 0 | 0 | 1.0 | 0 | 0 |
| | Alpaca | 0.54 | 0.13 | 0.32 | 0.42 | 0.45 | 0.12 | 0.43 | 0.26 | 0.30 |

Table 6: Predicted scoring of various parsers and DCQA questions using the best metrics. Reference-free: GPT4-Ans (Comp.), GPT-Cls-zs (Givn.), GPT4-Scr (Relv.). Reference-based: BERTScore (Comp.), GPT-4 (Givn.), GPT-4 (Relv.).

| Metric | Corr. | Metric | Corr. |
| --- | --- | --- | --- |
| BLEU-1 | 0.12 | METEOR | 0.21 |
| BERTScore | 0.32 | QSTS | 0.23 |
| ROUGE | 0.19 | GPT4-Scr | **0.57** |

Table 7: Spearman rank correlation coefficients of automatic metrics with annotated QUD similarity. The correlation values are significantly higher than 0 ($p < 0.05$).

ble 11). The reference-based metric values are mapped to labels using the same mapping function mechanism described in Section 5.1.

### 5.3 Results and Analysis

Results are shown in Tables 4 (reference-free) and 5 (reference-based), respectively. For reference, we report a **random baseline** that samples according to the distribution of labels in QUDEVAL.

Reference-free metrics score substantially better than reference-based ones, which are only slightly better than random. This questions the validity of using reference-based metrics for QUD parser evaluation, which we analyze further.

Notably, only GPT-Scr for the Relevance criterion is close to the human upper bound (Table 2). The performance on minority classes (see Table 1 for class frequencies) are especially low. GPT-Scr turned out to be often the best metric across different criteria; however, we point out caveats with its usage later in analysis.

**Do these metrics rank systems correctly?** It is conceivable that, despite these metrics being imperfect detectors of errors, they might still give reliable aggregate judgments about systems. To visualize the impact of applying these metrics to score systems, we show the *predicted* system performance using the best metrics for each category

is no entity overlap between $\hat{Q}$ and $Q$. We found this too restrictive for QUDs that are less factoid and entity-centric; thus our variant uses the arithmetic mean instead.

in Table 6. In general, *the metrics do not adequately capture system-level ordering determined by human judges in Table 1*. GPT-Scr is used for the Givenness and Anchor Relevance criterion; in both, it scores ChatGPT-generated QUDs much higher than it should, and even higher than human QUDs. This confirms the bias that GPT can favor itself when being used as an evaluation tool (Liu et al., 2023), although we did not include GPT-4 generated QUDs in the evaluation data per se.

**Do reference-based metrics capture similarity to annotated QUDs?** One hypothesis is that standard reference-based metrics are not actually capturing similarity with ground-truth QUDs. To check this, our annotators rated a subset of 200 pairs of $(\hat{Q}, Q)$ for similarity (detailed in Appendix C). This annotation allows us to examine label distributions stratified by how similar generated QUDs are to the reference QUDs (Appendix D, Figure 9). While questions similar to the reference QUDs tend to score better, our metrics do not clearly capture question similarity well (Table 7). Furthermore, even knowing similarity may not be enough to evaluate all cases of QUD, particularly when there are many possible QUDs that models can generate.

**Do reference-based metrics capture QUD quality?** Appendix Figures 7, 8 show the score distributions given by the best reference-based metrics split by ground-truth annotator label. For both METEOR and BERTScore, the score distributions are nearly identical across the categories, suggesting that it is tricky if not infeasible to tune a good threshold for such automatic metrics to map to our evaluation framework. This observation aligns with results from Table 5 where the metrics perform only slightly better than the random baseline. Since QUDs reflect discourse interpretations and hence are subjective, *we believe this is strong evidence*

*that imperfect standard reference-based methods should not be used for QUD evaluation.*

## 5.4 Discussion and Future Directions

As we established previously, QUD has utility for downstream generation tasks (Narayan et al., 2023; Newman et al., 2023; Wu et al., 2023). These recent lines of work indicate that question generation and answering is becoming more and more prominent as a way of handling discourse representation, despite a lack of rigorous evaluation of the generated questions themselves. This paper fills this gap by establishing a benchmark to evaluate automatic metrics, which are annotated by trained linguists. The standard metrics evaluated in this section show that we are far from having a reliable automatic metric for QUD evaluation; future work can iterate on more sophisticated methods to derive metrics, e.g., via supervised learning, iterative prompting, or chain-of-thought prompting.

## 6 Conclusion

We present a dataset, QUDEVAL, for evaluating contextual question generation in the context of QUD discourse parsing. QUDEVAL implements prior theoretical evaluation criteria for QUD, operationalized as four criteria for which expert linguistics annotators give high-quality judgments. This dataset sheds light on the divergent performance of existing systems, and enables future work on developing stronger automated metrics for evaluating QUD discourse parsing.

## Limitations

QUDEVAL evaluates each QUD edge independently thus does not take into account the relationship *between* questions. While full, hierarchical QUD trees constrain the answer of a QUD to entail the answers of its descendants (Roberts, 2012), a QUD dependency tree does not inherit this property (Ko et al., 2023). Thus we leave for future work to explore potential global constraints.

QUDEVAL is collected by trained linguistics students. We are yet to explore reliable ways to scale this to a crowdsourcing platform with the hopes of speeding up data collection. QUDEVAL is also limited to English newswire data; future work should explore other languages and genres.

The metrics we evaluated are baseline metrics. As pointed out in Section 5.4, this leaves headroom for future metric development.

## Acknowledgements

Special thanks to Kathryn Kazanas, Keziah Reina and Karim Villaescusa F. for providing data annotation for this project. This research was partially supported by National Science Foundation (NSF) grant IIS-2145479. We acknowledge the Texas Advanced Computing Center (TACC)[6] at UT Austin for many of the results within this paper.

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

# A Annotation Interface

Figure 4 shows the full annotator instruction. In Figure 5, we present our system UI for QUD evaluation. The answer sentence, anchor sentence, and prior context are highlighted. Questions to be evaluated are bolded, with other information displayed in gray.

Each criterion has selectable options, and there is also a text box for annotators to provide additional comments on specific errors. When the evaluation for a QUD is completed, the corresponding block is marked in light blue to track progress. If a question does not make sense, all criteria are marked as "skipped" and the block is also highlighted in light blue.

After submitting their answers, annotators can view the feedback in the table shown in Figure 6 to check their annotations and they can also copy the survey code directly.

# B Implementation Details for Reference-free Metrics

(1) *GPT-Scr* (**Answer Compatibility**): We prompted GPT-4 to generate a score between 1 to 100 for how well $S_a$ answers $\hat{Q}$. Our mapping uses tuned thresholds on the validation set: a score of over 80 was mapped to 'Direct and explicit', between 60 and 80 was mapped to 'Unfocused' and a score lower than 80 was mapped to 'Not answered'. Our best prompt is present in Table 9.

(2) *Hou (2021)* (**Givenness**): We found that running the full mention detection process is prohibitively slow, thus we extracted the maximum NP and their heads to speed up mention detection.

We differentiate answer leakage vs. hallucinations in 'new' mentions with the same mechanism as our rule-based metric: we check if content words (lemmatized; also excluding question (wh-) words, pronouns and names) in $\hat{Q}$ are present in $C_{\hat{Q}}$ and $S_a$. If there are content words absent in $C_{\hat{Q}}$ but found in $S_a$, $\hat{Q}$ is labeled as 'answer leakage'. If there are content words absent in both, they are deemed as 'hallucination'. Otherwise, we label $\hat{Q}$ as 'no new concepts'.

(3) *GPT-Scr* (**Anchor Relevance**): The prompt is shown in Table 16. A score of 80 and above

is mapped to 'fully grounded', 80-20 to 'partially grounded', and below 20 to 'ungrounded'.

(4) **BLEU1-sim**: If the BLEU-1 score between the anchor and the question is greater than 0.05, we consider the QUD fully grounded in its predicted anchor. If the score falls between 0.01 and 0.05, we classify it as partially grounded. Otherwise, it is considered not grounded.

## C  Question Similarity Annotation

The annotators were given pairs of $(\hat{Q}, Q)$ and $S_a$, as well as the full article context (66, 66, and 67 questions generated from Ko et al. (2023), ChatGPT and Alpaca respectively). Each question pair consisted of a human-generated question and a model-generated question for the same sentence within an article.

Similar to the setup in DCQA, scores between 1 and 5 were given based on the extent of similarity, 5 being the score given to identical or paraphrased questions. Each question is annotated by 2 annotators and their average score is considered for our analysis. The inter-annotator correlation is 0.728 and Krippendorff's $\alpha$ for the annotators is 0.687.

## D  Trends on Question Similarity and QUD Quality

In Figures 9a and 9b, we visualize the categorization of Answer Compatibility and Givenness for the 3 QUD parsers across 5 question similarity intervals. We can observe that for Answer Compatibility, instances with high similarity scores tend to mostly have $S_a$ that explicitly answers $\hat{Q}$. As the similarity score decreases, a greater percentage of instances are either Unfocused or Not answered. For ChatGPT however, across all similarity score intervals, the focus of $S_a$ answers $\hat{Q}$ most frequently, owing to the verbose nature of its generated questions which leak the answer concepts.

A similar trend can be observed for Givenness, where the percentage of answer leakage and hallucination increases with the decrease in similarity score. However, across the 3 parsers, even the most similar $(\hat{Q}, Q)$ pairs show answer leakage. Since Givenness is also an evaluation of $S_{\hat{k}}$, a plausible cause for such an observation is a bad $\hat{k}$ prediction. To further investigate this, we analyze the instances with the highest similarity scores (between 4 and 5) with the parser's $S_{\hat{k}}$ same as gold $S_{\hat{k}}$, generated for the same $S_a$. It was observed that barring ChatGPT (which showed answer leakage for similarity score

= 5), all instances were categorized as "No new concepts".

These observations indicate that the human-annotated in-context similarity scores somewhat correlate with both criteria experimented on here.

# Instructions

This task investigates the capability of AI to generate good reading comprehension questions.
Imagine you are reading the following article where the first few sentences go like this:

**1 The stock market's woes spooked currency traders but prompted a quiet little party among bond investors.**
**2 Prices of long-term Treasury bonds moved inversely to the stock market as investors sought safety amid growing evidence the economy is weakening.**

At this point, you may want to ask the question: *How much did the prices of long-term Treasury bonds increase?* This question is answered in sentence 7:

**7 At its strongest, the Treasury's benchmark 30-year bond rose more than a point, or more than $10 for each $1,000 face amount.**

Indeed, we can view sentences in a document as answers to questions that come from the readers, it's just that these questions are not explicitly stated. The goal of the AI model is to **recover** these questions and ask them explicitly. Having these questions will be helpful for readers who have trouble understanding the document.

Clearly, the question above is a good one: it is anchored in the second sentence, not overly generic, and sentence 7 is an answer that fits right in. But not all AI-generated questions are as good; consider this one: *How many points did Treasury's benchmark 30-year bonds increase?* This question is still answered by sentence 7, but it is odd to imagine a reader coming up with this question from sentence 2, because the notion of "points" (and maybe even "30-year") has not been introduced and only very expert readers would be familiar with this! On the other hand, we also don't want questions like, *What happened next?* This question is way too generic to be useful for a puzzled reader trying to understand the text.

To define a few terms: we call sentence 2 **the anchor sentence**, i.e., where the questions could reasonably arise from, and sentence 7 is the **answer sentence**. **Sentences before the anchor** form the question context. Sentences between the anchor and answer are faded as they should not provide extra information for the question.

We invite you to judge the quality of the questions based on the following criteria:

1. **Is this question make sense?**
   a. Yes
   b. No - Examples of really bad questions that don't make sense are:
      • Bad language (grammar errors, incomplete)
      • Irrelevant to the article (e.g., peace talks in article vs. climate change in question)
      • Straight off contradiction or too much hallucination
   **If the answer to this is "NO", then the rest of the annotation should be skipped.**

2. **Does the "answer sentence" actually answer this question?** There are three possibilities here:
   a. Explicit and direct answer: the "answer sentence"'s main content answers this question. Note that if the answer is provided in multiple sentences and the "answer sentence" is only one of them, that is ok.
   b. Unfocused answer: the "answer sentence" isn't really about answering the question, but some parts of the answer sentence answer the question or the answer could be inferred. E.g., Sentence 2 isn't about the evidence of a weakening economy, hence it is an "unfocused" answer to the question "How is the economy?" anchored in sentence 1.
   c. Not-an-answer: the "answer sentence" is not answering the question.

3. **Does the question contain new concepts that a reader would be hard to come up with? (By "new concepts", we mean concepts that cannot be easily inferred by world knowledge from existing ones).** There are several possibilities here as well:
   a. This question does not contain new concepts.
   b. Answer leakage: The question contains new concepts that are in the answer sentence AND not in the anchor sentence or the question context. "How many points did Treasury's benchmark 30-year bonds increase?" would be a question that leaks new concepts in the answer sentence: "30-year bonds" would not be a concept that most people would automatically jump to.
   c. Hallucination: The question contains new concepts. This includes:
      • Concepts not in the article.
      • The question contains new concepts that are not in the anchor sentence or the question context, but can be found later in the document.

4. **Is this question grounded well in the anchor sentence?** Here we ask you to judge from a scale:
   a. The question is fully grounded in the anchor sentence
   b. Some parts of the question is grounded in the anchor sentence
   c. The question is not at all grounded in the anchor sentence
   As an example, given the anchor "U.S. exports of nuclear material cannot be adequately traced from country to country, according to a congressional report", the question "What does the report say is the reason for the export ban" is only partially grounded, because although "the report" was mentioned, "export ban" was not.

5. **Extra issues**
   a. Redundant question: question already answered in anchor
   b. Question too generic to be informative/can be asked anywhere in the document
   c. Anything else

**Once all criteria for one question are answered, the question will be marked lightblue. This way you can know which question you haven't finished yet.**

Figure 4: Annotation instructions for QUDEVAL.

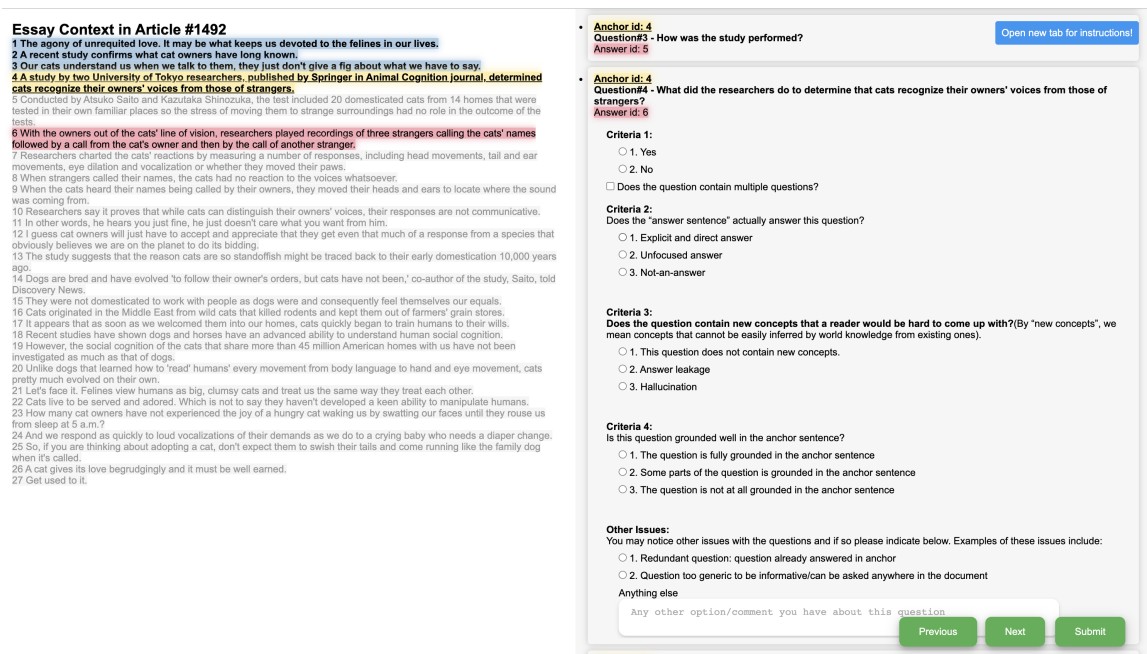

Figure 5: Annotation system UI.

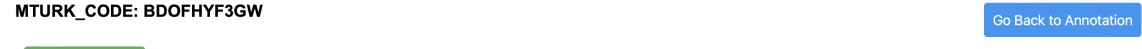

Go Back to Annotation

Copy to Clipboard

| Questions | Criteria 1 | Criteria 2 | Criteria 3 | Criteria 4 | Other Issues | Other Issues(typing) | Multi-question |
|-----------|-----------|-----------|-----------|-----------|--------------|----------------------|----------------|
| Question 0 | 1 | 3 | 2 | 2 | None | | False |
| Question 2 | 1 | 1 | 1 | 1 | None | | False |
| Question 3 | 1 | 1 | 1 | 1 | None | | False |
| Question 4 | 1 | 1 | 1 | 1 | None | | False |
| Question 6 | 1 | 1 | 1 | 1 | None | | False |
| Question 9 | 1 | 1 | 1 | 1 | None | | False |
| Question 10 | No | skipped | skipped | skipped | skipped | | False |
| Question 11 | 1 | 1 | 2 | 2 | None | | False |
| Question 15 | No | skipped | skipped | skipped | skipped | | False |
| Question 16 | 1 | 3 | 1 | 1 | None | | False |

Figure 6: Annotation system feedback.

**Step 1. Question Generation**

Examples for this question generation are:

**Context**: The stock market's woes spooked currency traders but prompted a quiet little party among bond investors. Prices of long-term Treasury bonds moved inversely to the stock market as investors sought safety amid growing evidence the economy is weakening. But the shaky economic outlook and the volatile stock market forced the dollar lower against major currencies. The bond market got an early boost from the opening-hour sell-off in stocks. That rout was triggered by UAL Corp.'s announcement late Monday that the proposed management-labor buy-out had collapsed. The 80-point decline in the Dow Jones Industrial Average during the morning trading session touched off a flight to safety that saw investors shifting assets from stocks to Treasury bonds.

**Target Answer**: At its strongest, the Treasury's benchmark 30-year bond rose more than a point, or more than $10 for each $1,000 face amount.

**Question**: How much did the prices of long-term Treasury bonds increase?

...[3 more in-context examples]

By reading the context, generate a question that indicates how the Target Answer elaborates on earlier sentences. The Target Answer given should be the answer to the generated question. The question should reflect the main purpose of the Target Answer. It should not use information first introduced in the Target Answer and shouldn't copy-paste phrases newly introduced in the Target Answer.

**Step 2. Anchor Selection**

**Context:** ...[same as above]

**Question:** ...[same as above]

**Anchor Sentence:** Prices of long-term Treasury bonds moved inversely to the stock market as investors sought safety amid growing evidence the economy is weakening

...[3 more in-context examples]

By reading the Context, pick a sentence from the Context such that the above Question arises from it. An Anchor Sentence is a sentence from the Context that the Question is most related to. The words and concepts from the Anchor Sentence are used to generate the Question.The Target Answer cannot be the Anchor Sentence.

Table 8: Few-shot prompt for QUD generation (with example article and answer sentence)

---

**article**: FORT LAUDERDALE, Fla. - Researchers are looking to the sun to give hunted and overfished sharks a new ray of hope. Using a special solar-powered tag, marine scientists now can study a shark's movements for up to two years by way of data beamed to satellites. Previously, researchers relied on tags that ran on batteries and sometimes died before all the information could be transmitted. The new tags are like a smartphone for marine animals,' said Marco Flagg, CEO of Desert Star, a Marina, Calif., company that offers solar devices.'Just like smartphones, the tags have many sensors and communication capability.The Guy Harvey Research Institute, based in Dania Beach, Fla., is looking to use solar tags to track certain species of fierce fish, including tigers, makos, hammerheads, oceanic white tip and sand sharks.The goal is to better understand their migratory patterns and ultimately keep their population healthy.Sharks are critical to the overall balance of ocean ecosystems, but commercial fishermen catch them by the millions for their fins, cartilage and meat.'We've learned a lot from tagging sharks, not least of which is that they are highly migratory,' said Antonio Fins, executive director of the Guy Harvey Ocean Foundation, which supports the institute.'They are not American sharks or Bahamian sharks or Mexican sharks.They don't know borders or nationalities.

**question**: Why are researchers studying sharks and using solar-powered tags to track their movements?

**answer**: Sharks are critical to the overall balance of ocean ecosystems, but commercial fishermen catch them by the millions for their fins, cartilage and meat.

**score**:

...

For the above article, give a score between 1 to 100 for how well the answer actually answers the question.

Table 9: GPT-Scr prompt for Answer Compatibility (with example article, question and answer)

---

**article**: FORT LAUDERDALE, Fla. - Researchers are looking to the sun to give hunted and overfished sharks a new ray of hope. Using a special solar-powered tag, marine scientists now can study a shark's movements for up to two years by way of data beamed to satellites. Previously, researchers relied on tags that ran on batteries and sometimes died before all the information could be transmitted. The new tags are like a smartphone for marine animals,' said Marco Flagg, CEO of Desert Star, a Marina, Calif., company that offers solar devices.'Just like smartphones, the tags have many sensors and communication capability.The Guy Harvey Research Institute, based in Dania Beach, Fla., is looking to use solar tags to track certain species of fierce fish, including tigers, makos, hammerheads, oceanic white tip and sand sharks.The goal is to better understand their migratory patterns and ultimately keep their population healthy.Sharks are critical to the overall balance of ocean ecosystems, but commercial fishermen catch them by the millions for their fins, cartilage and meat.'We've learned a lot from tagging sharks, not least of which is that they are highly migratory,' said Antonio Fins, executive director of the Guy Harvey Ocean Foundation, which supports the institute.'They are not American sharks or Bahamian sharks or Mexican sharks.They don't know borders or nationalities.

Which sentence in the article is closest to the sentence: 'Researchers are studying the movements of sharks using a special solar-powered tag that can transmit data to satellites for up to two years. '

**A**:

Table 10: GPT-Ans prompt for Answer Compatibility (with example article and GPT-4 generated answer)

**Context**: The Justice Department is in the process of trying to gain control over a law that federal Judge David Sentelle recently called a 'monster.' Needless to say, he was talking about RICO.With its recently revised guidelines for RICO, Justice makes it clear that the law currently holds too many incentives for abuse by prosecutors.The text of the 'new policy' guidelines from the Criminal Division are reprinted nearby.They strongly suggest that Justice's prosecutions of Drexel Burnham Lambert, Michael Milken and Princeton/Newport violated notions of fundamental fairness.Justice is attempting to avoid a replay of these tactics.This amounts to an extraordinary repudiation of the tenure of New York mayoral candidate and former U.S. Attorney Rudolph Giuliani, who was more inclined to gathering scalps than understanding markets.The new guidelines limit the pretrial forfeitures of assets of RICOed defendants and their investors, clients, bankers and others.This follows earlier new guidelines from the Tax Division prohibiting Princeton/Newport-like tax cases from masquerading as RICO cases.
**Reference Question**: What is the rationale for limiting the pretrial forfeitures?
**Candidate Question**: In what way are forfeitures limited now?
**Score**:

...

Given the Context, score the above Candidate Question for similarity with respect to the Reference Question on a continuous scale from 1 to 5, where a score of 1 means 'no similarity' and a score of 5 means 'similar intent and phrasing'

Table 11: Prompt for assessing similarity between $(\hat{Q}, Q)$ (with example context, reference and candidate question)

**Context**: 1 CHARLESTON, W.Va. - Downtown businesses and restaurants began to reopen after water was declared safe to drink in portions of West Virginia's capital, but life has yet to return to normal for most of the 300,000 people who haven't been able to use running water in the five days since a chemical spill. 2 It could still be days before everyone in the Charleston metropolitan area is cleared to use water, though officials say the water in certain designated areas was safe to drink and wash with as long as people flushed out their systems. 3 They cautioned that the water may still have a slight licorice-type odor, raising the anxieties of some who believed it was still contaminated. 4 'I wouldn't drink it for a while. I'm skeptical about it,' said Wanda Blake, a cashier in the electronics section of a Charleston Kmart who fears she was exposed to the tainted water before she got word of the spill.
**Question**: How widespread were the effects of the spill?
**Answer**: Thursday's spill affected 100,000 customers in a nine-county area, or about 300,000 people in all. Does the question contain new concepts that a reader would be hard to come up with? (By "new concepts", I mean concepts that cannot be easily inferred by world knowledge from existing ones). There are several possibilities here as well: This question does not contain new concepts. Answer leakage: The question contains new concepts that are in the answer sentence AND not in the context. Hallucination: The question contains new concepts. This includes: Concepts not in the article. The question contains new concepts that are not in the context, but can be found later in the document.

Given the Context, Question, and Answer, select one of the following options on the basis of your understanding of the instructions.
1: No new concepts
2: Answer leakage
3: Hallucination

Table 12: GPT-Cls-zs prompt for Givenness (with example context, question and answer).

Here are a few examples for all cases:
**Example 1:**
**Context:** 1 U.S. exports of nuclear material cannot be adequately traced from country to country, according to a congressional report.
**Question:** What does the report say is the reason for the export ban?
**Answer Sentence:** The report says hundreds of tons of plutonium and highly enriched uranium have accumulated worldwide, mostly from nuclear power generation.
**Selected option:**
[3: Hallucination]
...[5 more in-context examples; each option has two examples]

Does the question contain new concepts that a reader would be hard to come up with? (By "new concepts", I mean concepts that cannot be easily inferred by world knowledge from existing ones). There are several possibilities here as well: This question does not contain new concepts. Answer leakage: The question contains new concepts that are in the answer sentence AND not in the context. Hallucination: The question contains new concepts. This includes: Concepts not in the article. The question contains new concepts that are not in the context, but can be found later in the document.

Given the Context, Question, and Answer, select one of the following options on the basis of your understanding of the instructions.
1: No new concepts
2: Answer leakage
3: Hallucination

Table 13: GPT-Cls-fs prompt for Givenness

**Question**: How widespread were the effects of the spill?
**Anchor Sentence**: 'I know I've ingested it'. By Tuesday morning, officials had given the green light to about 35 percent of West Virginia American Water's customers.
Does the question contain new concepts that a reader would be hard to come up with? (By "new concepts", I mean concepts that cannot be easily inferred by world knowledge from existing ones). There are several possibilities here as well: This question does not contain new concepts. Answer leakage: The question contains new concepts that are in the answer sentence AND not in the context. Hallucination: The question contains new concepts. This includes: Concepts not in the article. The question contains new concepts that are not in the context, but can be found later in the document.

Is the question well-grounded in the anchor sentence? Please evaluate using the following scale:
1: The question is fully grounded in the anchor sentence.
2: Some parts of the question are grounded in the anchor sentence.
3: The question is not grounded at all in the anchor sentence.

Based on the question and the anchor, please choose one of the above options. If the question refers to the same entity as the anchor, we consider the question to be grounded.

Table 14: GPT-Cls-zs prompt for Anchor Relevance (with example question and anchor sentence)

Here are a few examples for all cases:
**Example 1:**
**Question:** What do lawmakers think about this issue?
**Anchor Sentence:** U.S. exports of nuclear material cannot be adequately traced from country to country, according to a congressional report.
**Result:** [1: The question is fully grounded in the anchor sentence.]
...[5 more in-context examples; each option has two examples]

Is the question well-grounded in the anchor sentence? Please evaluate using the following scale:

1: The question is fully grounded in the anchor sentence. 2: Some parts of the question are grounded in the anchor sentence. 3: The question is not grounded at all in the anchor sentence.
Based on the question and the anchor, please choose one of the above options. If the question refers to the same entity as the anchor, we consider the question to be grounded.

Table 15: GPT-Cls-fs prompt for Anchor Relevance

**Question:** What do foreign policy experts say about the issue?
**Anchor Sentence:** U.S. exports of nuclear material cannot be adequately traced from country to country, according to a congressional report.
Based on the question and the anchor, give a score between 1 to 100 for how confident you are about the question is grounded in anchor sentence. If the question refers to the same entity as the anchor, we consider the question to be grounded.

Table 16: GPT-Scr prompt for Anchor Relevance (with example question and answer sentence)

| Question |
| --- |
| What is the main objective of Clinton in forging a wedge between Milosevic and the Serbs in |
| What will happen to owners who cannot distinguish their owners' voices? |

Table 17: Examples of questions that failed the Language Quality criterion.

| Question | Answer |
| --- | --- |
| What happened after he took the students to Poland? | But he wasn't prepared for the uproar that followed. |
| What was the reason for the scheduled resumption of peace talks in Ingushetia? | The scheduled resumption of talks in the town of Sleptsovsk came two days after agreement on a limited cease-fire, calling for both sides to stop using heavy artillery Tuesday. |

Table 18: Examples where the answer sentence is an "unfocused" label for Answer Compatibility. In the first example, the answer does not directly talk about an event. In the second example, the answer is not the focus of the sentence. This is because the cease fire alone doesn't seem to be the reason for the resumption of peace talks

**Anchor (S1):** SEATTLE - There's little lyrical language to be found in the most recent international report on climate change.
**Question:** What is the essence of the IPCC report?
**Answer:** So when a bad cold kept him in the house one weekend, Johnson decided to distill the report to its essence via a centuries-old Japanese art form: haiku.

**Anchor (S1):** Fundamental freedoms are lagging behind rapid economic growth in Vietnam, according to a new U.N. report.
**Question** What is the purpose of the U.N. Human Rights Ombudsman's visit?
**Answer** The group visited Vietnam for one week in October last year.

Table 19: Examples where $\hat{Q}$ is labeled "hallucination" for Givenness. Both anchor sentences are identified to be the first sentence. In the first example, IPCC is not in the common ground (S1) nor in $S_a$, and inferring that the international report was written by IPCC is quite a leap; in the second sentence, "U.N. Human Rights Ombudsman's" is also unknown to the reader.

**Answer Compatibility**
**Annotation1:** not answered, **Annotation2:** not answered, **Annotation3:** unfocused
**Anchor Sentence:** But the technological sleuthing it took a group of Carnegie Mellon University students and alumni to recover and preserve some digital images apparently created and stored by Andy Warhol on old-school floppy computer disks nearly 30 years ago is a tale worth telling.
*Question: What was unique about these digital images?*
**Answer Sentence:** Those three images of an altered Botticelli's 'Venus,' a Warhol self-portrait, and a Campbell's soup can - of 28 that were found on the disks - were enough to excite Warhol fanatics around the world over the possibility that something - anything - new by the King of Pop Art had been revealed.

**Anchor Relevance**
**Annotation1:** fully grounded, **Annotation2:** partially grounded, **Annotation3:** partially grounded
**Anchor Sentence:** Every spring, from Florida to New Jersey, crabs that look more like fossils than a postcard for passion make their way ashore by the thousands when the moon is bright to lay millions of eggs that provide critical food for migrating shorebirds.
**Question:** What happened to their numbers?
**Answer Sentence:** But in the 1990s, their numbers began falling.

Table 20: Examples where annotators disagree. In the first example, the Answer Sentence mainly focuses on the excitement around Warhol fanatics on the revelation of the three digital images and what they were. Although it mentions that this maybe new work, suggesting it might be unique (quite subjective), it doesn't seem to be the main focus of the Answer Sentence, justifying the 'unfocused' annotation. For the second example, no concept in the question is discourse-new and wondering about what happened to the numbers is natural to some readers, justifying the 'fully grounded' annotation of one of the readers.

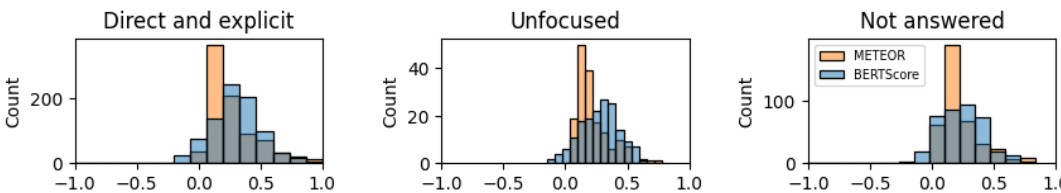

Figure 7: Distribution of automatic metrics for Answer Compatibility.

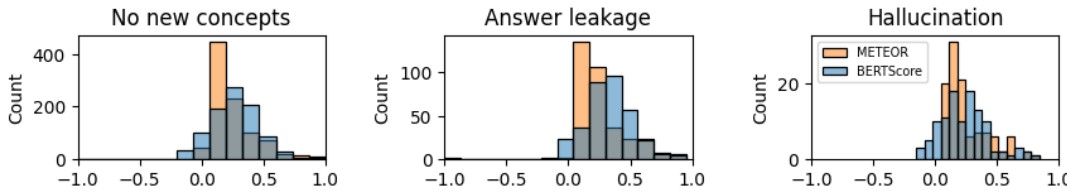

Figure 8: Distribution of automatic metrics for Givenness.

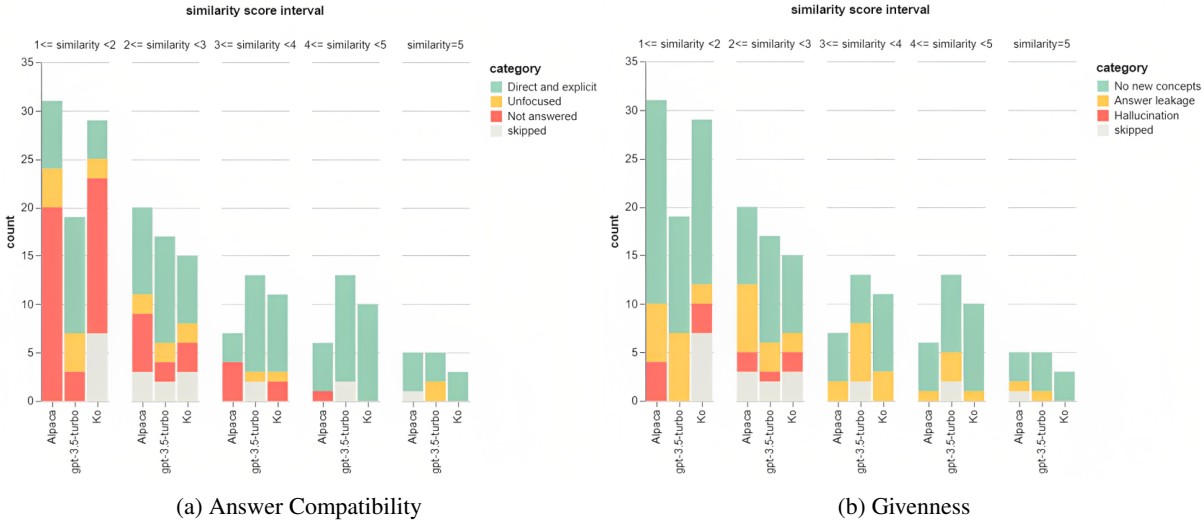

(a) Answer Compatibility

(b) Givenness

Figure 9: Categorization of Answer Compatibility, Givenness across 5 similarity score intervals