# OpenReview forum: "QUDeval: The Evaluation of Questions Under Discussion Discourse Parsing"
_EMNLP/2023/Conference — EMNLP 2023 Main_

### Official Review · Reviewer_ttgK · 2023-08-05

**Soundness:** 3

**Excitement:**

4: Strong: This paper deepens the understanding of some phenomenon or lowers the barriers to an existing research direction.

**Paper Topic And Main Contributions:**

This paper introduces an evaluation framework for the Questions Under Discussion (QUD) task. The task itself involves generating a question from an article that is answered by a sentence within the article, and determining the anchor sentence within the article which inspired the generated question. The first contribution of the paper is a set of evaluation criteria for this task. The criteria are inspired by an error analysis of existing QUD systems and by evaluation criteria used in previous studies. The second contribution of the paper is the application of these criteria to the output of 3 QUD systems (one fine-tuned, and two based on LLM prompting) and to the dev set of an existing QUD dataset (DCQA) whose questions have been manually generated by crowdsourcing. The system outputs and the human-generated questions are annotated by a single trained annotator according to the evaluation criteria - this data will be released. A third contribution is the analysis of the system and human performance on the task - the main findings here are that the ChatGPT system produces the questions most compatible with the answer sentences but is prone to answer leakage (where the question is grounded in information only available in the answer sentence), and that the system fine-tuned on the DCQA dataset is the most prone to hallucination but is also best at predicting the anchor sentence. The Alpaca system has a tendency to repeat the same questions, given the same article as input. The human-generated questions are evaluated highest overall, although they are not deemed as compatible with the answer sentence as the ChatGPT-generated questions. Finally, the last contribution is the use of the new dataset to evaluate automatic metrics (reference-free and reference-based). The results show that there is much room for improvement for these metrics, particularly the reference-based ones.


**Questions For The Authors:**

Does the order of the subtasks matter? Could an anchor sentence be identified first, followed by question generation?

Can there be more than one anchor sentence for a given question and answer?

Can the first sentence in the document be considered an answer sentence (see def in 206-208)?




**Reasons To Accept:**

* A new and useful dataset
* A new set of well justified evaluation criteria which will likely be used by other researchers working on this task
* A thorough set of experiments with some findings that reveal avenues for future progress on this task
* A clearly written paper



**Reasons To Reject:**

* Motivation - the introduction doesn’t do a good enough job in explaining why this is a useful task. The role of QUD in the applications listed in 43-46 need to be spelled out more.

* Annotation quality - the inter-annotator agreement isn’t particularly high and some of the examples in the appendix raised a few questions for me (see Annotation section below)

* Relatively minor issues in presentation which nonetheless should have been caught before submission - see below



**Reproducibility:**

3: Could reproduce the results with some difficulty. The settings of parameters are underspecified or subjectively determined; the training/evaluation data are not widely available.

**Reviewer Confidence:**

4: Quite sure. I tried to check the important points carefully. It's unlikely, though conceivable, that I missed something that should affect my ratings.

**Typos Grammar Style And Presentation Improvements:**

Clarification
* 169-171: say why these criteria are insufficient
* 381 - clarify that these 150 are human-generated
* 406 - “Qualitative Analysis” based on what? Provide more detail.
* Section 5.2 - clarify where the reference questions come from. The crowd-sourced DCQA questions?

Annotation
* Line 300 - the term ‘’irrelevant” is too strong here. The question is clearly not irrelevant to the anchor sentence since both refer to nuclear materials.
* Table 19: the second example doesn’t seem like an example of “unfocused”. The answer sentence implies that the resumption of peace talks came as a result of the cease-fire. What is the focus if not this?
* Table 20: there’s a clear difference between the two examples of hallucination. In the first example, the IPCC report is in fact mentioned in the first sentence - it’s just that the IPCC acronym is not used.
* The annotator disagreement examples given in the Appendix (Table 21) should be discussed. The second one (anchor relevance) looks like an error on the part of one annotator, although on lines 376-377 it says that no instance of task misinterpretation on the part of an annotator was found? How could the question be “fully grounded” in the anchor sentence in this example?
* 578: do people really agree with “high reliability”? IAA is only moderate

Presentation Details
* Table 5: is the heading wrong in the last section? I assume these are the Anchor Relevance results?
* There are a lot of informative examples in the appendix and it’s distracting to have to switch back and forth. Consider moving a few to the main paper for a nine-page camera-ready version.
* Grammar error in the instructions to the annotators (on the criteria related to language quality)
* Table 1: No col not needed for Lang criteria
* Table 4: the upper bound from Table 2 should be included somewhere. Maybe in the caption?

---

> ### Author Rebuttal · Authors · 2023-08-28
>
> _Re: motivation:_\
> Please also see the first point of our response to R2. We will incorporate this framing of the motivation into the paper when extra space is available.
>
> _Re: annotation:_\
> We address your concerns towards the end of this response. We believe that most of the concerns came from the lack of full document context rather than annotator error; we will address that by putting more context into the tables.
>
> _Re: Does the order of the subtasks matter? Could an anchor sentence be identified first, followed by question generation?_\
> That's a good question. Yes, it's possible. In fact, that's how Ko et al’s QUD parser operates (as indicated for R1, we will include a brief summary of their parser in the appendix). In GPT models and Alpaca, however, we found that the current order better guides those models, resulting in a superior QUD output.
>
> _Re: Can there be more than one anchor sentence for a given question and answer?_\
> It is both theoretically and operationally possible that more than one sentence satisfies the criteria of being an anchor. In practice, we see that this depends on varying levels of specificity of the question and the document context. Because of this, our evaluation framework takes into account this possibility by independently evaluating each QUD and its predicted anchor.
>
> _Re: Can the first sentence in the document be considered an answer sentence (see def in 206-208)?_\
> In our experiment, the first sentence is not considered an answer sentence. It is possible, however, to treat it as one if the document title is taken into account. As neither datasets nor parsers in prior work did so, we did not feel right to evaluate it as such.
>
> _Re: Line 300 - the term ‘’irrelevant” is too strong here. The question is clearly not irrelevant to the anchor sentence since both refer to nuclear materials:_\
> What we meant here is only scoped towards the specific requirement for anchor groundedness, i.e., concepts in the question needs to be hearer-old. We will clarify this. While we agree that both the anchor sentence and question mention nuclear materials, the context of usage is very different. The anchor sentence mentions what’s present or missing in the report, namely that the report talks about the export license of nuclear material. The question is centered around the restrictions on nuclear exports and transfer of nuclear material. The concept of “restrictions” is the main focus of the question which is irrelevant to the anchor sentence that provides more information on the contents of the report.
>
> _Re: Table 19: the second example doesn’t seem like an example of “unfocused”. The answer sentence implies that the resumption of peace talks came as a result of the cease-fire. What is the focus if not this?_\
> The annotators did provide their reasoning for the unfocused label (before we submitted the paper): given the article context, the annotator didn’t think that the cease fire alone is the reason for the resumption of peace talks; rather they believed that it should be the reason for the cease fire.
>
> _Re: Table 20: there’s a clear difference between the two examples of hallucination. In the first example, the IPCC report is in fact mentioned in the first sentence - it’s just that the IPCC acronym is not used._\
> We disagree. The anchor sentence just says "international report on climate change." It is quite a leap to infer that the international report was necessarily written by the IPCC ("Intergovernmental Panel on Climate Change") and not some other organization. In our view, most readers would not write this question unless they already had deep prior knowledge of this topic.
>
> _Re: The annotator disagreement examples given in the Appendix (Table 21) should be discussed. The second one (anchor relevance) looks like an error on the part of one annotator, although on lines 376-377 it says that no instance of task misinterpretation on the part of an annotator was found? How could the question be “fully grounded” in the anchor sentence in this example?_\
> We will happily discuss the disagreements. For this particular example, we disagree that this is not “fully grounded”: no concept in the question is discourse-new, and that wondering about what happened to the numbers (that were explicitly mentioned in the anchor) is natural to some readers (though there is a degree of subjectivity here).
>
> _Re: Line 578: do people really agree with “high reliability”? IAA is only moderate._\
> We apologize for the confusion that the reference to Table 2 makes. “High reliability” doesn’t necessarily mean high agreement, which isn’t true for most of the NLP tasks (e.g., coreference resolution, entailment, etc). Our annotators are linguists well-trained for this task, and we have been through multiple rounds of discussion about potential disagreements. We are highly confident about the quality of the annotation.
>
> We also thank the reviewer for the clarification and presentation suggestions; we appreciate them and will address/fix them!

---

### Official Review · Reviewer_WebN · 2023-08-09

**Soundness:** 4

**Excitement:**

2: Mediocre: This paper makes marginal contributions (vs non-contemporaneous work), so I would rather not see it in the conference.

**Paper Topic And Main Contributions:**

The paper proposes QUDeval, a data set containing automatically generated (implicit-question, anchor) pairs with human quality annotation. The goal of the paper is to provide a framework for evaluation of evaluation metrics for QUD parsing.

**Questions For The Authors:**

- There is little explanation on how the reference-based metrics are mapped to integers. Can you explain this?

- Fn2: What is the 'temperature', why is it set to 0.0 and not to some other value?

**Reasons To Accept:**

- The paper is well written and the annotation protocol of the resource is very carefully described. Therefore, I presume that the annotation is of good quality. I appreciate this.

- The human upper-bound seems to suggest that automatic metrics for QUD parsing (even the ones based on LLMs) have a long way to go to properly solve the task. If true, this paper can show a weakness of LLMs, which may valuable.

**Reasons To Reject:**

Even though the paper tries motivating the task/resource, I have some doubts on whether an evaluation benchmark for evaluation metrics of QUD is/will be a relevant resource for the community. One reason is that QUD parsing doesn't seem very active. Another reason is that it is not really clear what conclusions we can draw from evaluating a metric on QUDeval. If a purpose of the benchmark is to enable QUD parsing researchers to better select a metrics for scoring QUD parsers, then from the presented results I think they will have a hard time doing this, since the metrics seem to score so differently in different aspects. Maybe it is also not the automatic metrics that are the culprit, but the main problem is that there aren't multiple references?

**Reproducibility:**

4: Could mostly reproduce the results, but there may be some variation because of sample variance or minor variations in their interpretation of the protocol or method.

**Reviewer Confidence:**

3: Pretty sure, but there's a chance I missed something. Although I have a good feel for this area in general, I did not carefully check the paper's details, e.g., the math, experimental design, or novelty.

**Typos Grammar Style And Presentation Improvements:**

- 449: there seems to be an 'and' too much or some grammar issue going on.

- Macro F1 doesn't seem like a good choice to evaluate the prediction of an ordinary scale. Obviously, the score 4 fits much better to 5 than to 1. With a softer calculation, the metrics may perform better, also many of them do not naturally assign a category and require some (possibly messy) thresholding (on which I find too little information).

- 578: I don't think table 2 shows "high reliability" of agreement among human annotators.

---

> ### Author Rebuttal · Authors · 2023-08-28
>
> Thank you for your feedback!
>
> As stated in the paper, QUD parsing is an emerging area of research. The lack of prior computational work is not an indication for a lack of work in the future. Considering the history of RST parsing, it took decades following Mann & Thompson’s theoretical work for it to become widely used in the NLP community, even after the first corpora & parser led by Daniel Marcu. We believe that it is important for an emerging area to have a sound evaluation paradigm grounded in linguistic theory, and this work aims to achieve that. The need for rigorous evaluation is further manifested by existing work that uses QUD for generation while lacking an intrinsic evaluation for question generation (e.g., Narayan et al ArXiv 2022 now TACL 2023, Newman et al. 2023, etc).
>
> As for the fact that we cover a few different aspects: this is necessary, as linguistic theories point to the fact that all these criteria need to be satisfied _jointly_. Future work can explore combining them into a “single score”, but we emphasize the need for fine-grained evaluation that is one of the key insights in recent work that evaluates NLG system outputs.
>
> _Re: multiple reference:_\
> While we agree that multiple references would be very nice to have, we want to point out that for most high-entropy generation tasks, reference-based evaluation is inherently flawed given the high number of valid outputs that are semantically different; Section 5 brings up this point. Our stance is that reference-free metrics with sound linguistic principles are the best way forward for such tasks.
>
> _Re: explanation of mapping for reference-based metrics:_\
> Thanks a lot for catching this! The mapping is the same mechanism as reference-free, stated in lines 452–457; we will fix this.
>
> _Why temperature is 0.0:_\
> Temperature is a hyperparameter that controls the randomness of the generated output. We noticed that increasing the temperature for our setting resulted in the generated questions and anchors containing concepts which are outside the context provided. Hence it was set to 0. We will add this point as a footnote into the paper.
>
> _“Typos Grammar Style And Presentation Improvements”:_
>
> _Re: 449:_ thank you for catching this. We'll fix this.
>
> _Re: macro F1:_ thank you for the suggestion. We opted for a Macro F1, along with a per-class F1 for all categories, because the number of classes is small (3), and not all of the aspects we measure have an ordinal semantics associated with them. We can definitely add an MAE measure to those that have this property.
>
> _Re: line 578:_ We apologize for the confusion that the reference to Table 2 makes. “High reliability” doesn’t necessarily mean high agreement, which isn’t true for most of the NLP tasks (e.g., coreference resolution, entailment, etc). Our annotators are linguists well-trained for this task, and we have been through multiple rounds of discussion about potential disagreements. We are highly confident about the quality of the annotation.
>
> _References mentioned above:_ (both cited in the paper)
> - Narayan et al., Conditional Generation with a Question-Answering Blueprint, TACL 2023
> - Newman et al., A Controllable QA-based Framework for Decontextualization, ArXiv 2023

---

### Official Review · Reviewer_FkHr · 2023-08-12

**Soundness:** 3

**Excitement:**

4: Strong: This paper deepens the understanding of some phenomenon or lowers the barriers to an existing research direction.

**Paper Topic And Main Contributions:**

This paper presents a dataset, QUDEVAL, for evaluating question generation in the context of
QUD discourse parsing. QUDEVAL implements
prior theoretical evaluation criteria for QUD, for which expert linguistics annotators give high-quality judgments.  Using the annotations, the authors evaluate the performance of existing QUD parsers and their own models.
They further come up with automatic metrics for the assessment of automatically generated QUDs. The annotations
and evaluation protocol are made available (annotator instructions are provided in the appendix).

**Questions For The Authors:**

- The paper is very dense. It would have been nice to actually talk about the discourse parsing task in more detail, show
an example of a dependency tree (either in the main text or the appendix).

- Some more detail on Ko et al.'s parser would also have been useful.

- Regarding the modeling, have you considered to use previously generated QUDs as input to generate the question in the next time step?
This might prevent models from generating repeated questions/answers.

- Please explain why you generated a few QUDs from several documents rather a complete set from a lesser number.

- Table 1, are differences in parser performance statistically significant?

- Table 4, please report what metric you are using to evaluate the automatic metrics.

- Table 5, why are you using ROUGE? You do not care about recall. Also please explain what QSTS is.

- Table 7, do you mean Spearman's rho? Are are these correlations statistically significant?

**Reasons To Accept:**

QUD parsing is not well-known or established in the research community. The paper takes an important step
towards evaluating the output of QUD parsers, it proposes theoretically established criteria and shows that
the proposed annotation scheme leads to relatively high annotation agreement. The paper further
proposes automatic metrics for the evaluation of QUD output, demonstrating there is room for improvement.
It further provides background on QUD, and develops parsing models based on GPT.

**Reasons To Reject:**

I cannot find any reason to reject, the paper is interesting, a serious effort towards the automatic evaluation of QUDs.

**Reproducibility:**

3: Could reproduce the results with some difficulty. The settings of parameters are underspecified or subjectively determined; the training/evaluation data are not widely available.

**Reviewer Confidence:**

4: Quite sure. I tried to check the important points carefully. It's unlikely, though conceivable, that I missed something that should affect my ratings.

---

> ### Author Rebuttal · Authors · 2023-08-28
>
> Thank you for your feedback and the recognition of the merit of our work. Below we address “Questions for the authors”:
>
> _Re: example of a dependency tree and details of Ko et al.’s parser:_\
> We appreciate your suggestions and will definitely include more examples and a short summary of Ko et al’s parser in the appendix.
>
> _Re: using previously generated QUDs:_\
> This is an interesting suggestion! We focused more on evaluating parsers, so this was a bit outside the scope of what we were experimenting with here, but we agree that's a great direction for future work.
>
> _Re: Please explain why you generated a few QUDs from several documents rather a complete set from a lesser number:_\
> We think that the utility of having a more diverse coverage in terms of topics across documents is higher (note that since the annotation is high in cognitive load, evaluating full, long documents would further complicate this task).
>
> _Re: Table 1, are differences in parser performance statistically significant?_\
> We ran the Wilcoxon rank sum test between all pairs of parsers. Other than language quality, differences across most parser comparisons in table 1 are statistically significant (p<0.05). Language quality differences are not significant as all systems were able to generate fluent and sensible questions; the Ko et al model does not differ significantly from ChatGPT or Alpaca in terms of unfocused answers, nor with ChatGPT in terms of anchor relevance. We’ll add this into the table.
>
> _Table 4:_ we are using per-class F1 and macro-averaged F1. Although we mentioned the macro F1 in the column name, we overlooked that the per-class F1 was not specified there. We will add this to the caption; thanks for flagging this.
>
> _ROUGE:_ We reported ROUGE-F1 in the context of *reference-based* evaluation only, since  it is often used for the evaluation of QG tasks. There, recall does matter if we are measuring the similarity between the generated output and the reference input.
>
> _QSTS_ (Gollapalli and Ng, 2022) stands for Question-Sensitivity Text Similarity. It explicitly represents the question class and named entities present in a given question pair and combines them with dependency tree information and word embeddings to measure the semantic similarity between 2 questions. QSTS is a state-of-the-art reference-based metric for question generation evaluation, hence we included it. We will add a short description to the appendix.
>
> _Table 7_ indeed has the Spearman Rank Correlation coefficients for the automatic metrics like METEOR, ROUGE against the human annotated QuD similarity with all the values being statistically significant. We will clarify significance in the caption.

---

### Meta-Review · Area_Chair_WPgw · 2023-09-03

**Recommendation:** 4
**Confidence:** 4

**Metareview:**

This paper presents detailed evaluation criteria, new evaluation data, and an evaluation of multiple outputs for the QUD discourse parsing task, an emerging area of computational discourse modeling. Reviewers generally agree that the quality of the paper is high, that the level of detail is a strong point, and that there are novel contributions here, while their views on the excitement and potential of this work moving forward are more mixed. All reviewers assign positive soundness scores, and two reviewers give a strong excitement score. Considering the relatively unexplored state of QUD research at this point, I would like to see more papers in this area get a chance to develop the framework further, and especially the contribution of new data could be substantial at this point. If accepted, I would ideally like to see this presented as a poster to encourage discussion and allow interested researchers to find out more about the data and how they can use it to develop these ideas further.

---

### Decision · Program_Chairs · 2023-10-07

**Decision:**

Accept-Main

**Comment:**

This paper presents detailed evaluation criteria, new evaluation data, and an evaluation of multiple outputs for the QUD discourse parsing task, an emerging area of computational discourse modeling. Reviewers generally agree that the quality of the paper is high, that the level of detail is a strong point, and that there are novel contributions here, while their views on the excitement and potential of this work moving forward are more mixed. All reviewers assign positive soundness scores, and two reviewers give a strong excitement score. Considering the relatively unexplored state of QUD research at this point, I would like to see more papers in this area get a chance to develop the framework further, and especially the contribution of new data could be substantial at this point. If accepted, I would ideally like to see this presented as a poster to encourage discussion and allow interested researchers to find out more about the data and how they can use it to develop these ideas further.